# Crystalline Structure, Synthesis, Properties and Applications of Potassium Hexatitanate: A Review

**DOI:** 10.3390/ma12244132

**Published:** 2019-12-10

**Authors:** Patricia Ponce-Peña, Martha Poisot, Alicia Rodríguez-Pulido, María Azucena González-Lozano

**Affiliations:** 1Facultad de Ciencias Químicas, Universidad Juárez del Estado de Durango, Av. Veterinaria s/n, Circuito Universitario, C.P. 34120 Durango, Mexico; pponce@ujed.mx (P.P.-P.); investigacion_arp@hotmail.com (A.R.-P.); 2Instituto de Química Aplicada, Universidad del Papaloapan, Circuito Central 200, Parque Industrial, C.P. 68301 Tuxtepec, Oax., Mexico; mpoisot@unpa.edu.mx

**Keywords:** potassium hexatitanate, photocatalyst, reinforcing agent, mechanical properties, hydrogen production

## Abstract

Potassium hexatitanate (PHT) with chemical formula K_2_Ti_6_O_13_ has a tunnel structure formed by TiO_2_ octahedra sharing edges or corners and with the potassium atoms located in the tunnels. This material has attracted great interest in the areas of photocatalysis, reinforcement of materials, biomaterials, etc. This work summarizes a large number of studies about methods to prepare PHT since particle size can be modified from millimeter to nanometric scale according to the applied method. Likewise, the synthesis method has influenced the material properties as band-gap and the final mechanical performance of composites when the reinforcement is PHT. The knowing of synthesis, properties and applications of PHT is worthwhile for the design of new materials and for the development of new applications taking advantage of their inherent properties.

## 1. Introduction

Alkaline metal titanates with tunnel structure attract great attention both experimental and theoretical for model systems, for being fast ionic conductors, as well as for their high photocatalytic activities. The family of alkaline titanates is represented by the general formula A_2_On TiO_2_ (3 ≤ n ≤ 8 and A = K, Na; Li). All of them have tunnel or laminar structure, with each layer consisting of zigzag ribbons infinitely extended in “b” direction, in which the octahedra share the edges in a level, in linear groups of three units, and are linked with each other by the corners [1], being “n” for the titanates with layered structure and n/2 for the titanates with tunnel type. This type of structure is explained in detail later, specifically for the potassium titanates family.

The reports on the synthesis of different types of potassium titanates began in the 1950s with the work described in 1958 in the U.S. patents 2 833 620 and 2 841 470, by Gier et al. and Berry K.L., respectively. However, it was not until the 1980s that this kind of materials acquired greater importance due to the properties they present, and deeper research began on different methodologies to produce them and on their diverse applications, research that continues into the present. It should be mentioned that the majority of the potassium titanates obtained initially were synthesized by flux method and had a fiber-like morphology of millimetric and micrometric sizes [2,3,4,5].

The potassium titanates family comprises a series of compounds of general formula K_2_O*n*TiO_2_, with *n* equal to 1, 2, 4, 6 or 8, commonly called potassium -titanate, -dititanate, -tetratitanate, -hexatitanate and -octatitanate, respectively [2,3,4,6]. Moreover, the compounds KTiO_2_ (OH) [7,8], K_3_Ti_8_O_17_ [9] and K_4_Ti_3_O_8_ and K*_x_*Ti_8_O_16_ [7,10,11,12] have also been reported but not extensively investigated. All of them have been recognized as important functional materials for a wide range of applications, due to their excellent chemical, thermal, optical, mechanical, catalytic, photocatalytic and frictional properties [13,14,15,16]. These properties can be varied from one titanate to another with the change of *n* [17]. The crystalline structure of titanates is formed by layers of TiO_6_ octahedra joined by edges and/or corners, creating zigzag ribbons united by the corners, with K^+^ ions accommodated between them delivering a 3D network, enclosing tunnels [1,2,11,12,18].

In the thermodynamic analysis of the K_2_CO_3_-TiO_2_ system (Figure 1) carried out using the software HSC Chemistry 7.0 [19], considering 1 atm of pressure and temperatures from 25 to 1600 °C as simulating conditions, consistent with the chemical reactions 1, 2, 3 and 4, the results showed a negative Gibbs free energy (ΔG) is necessary to the formation of different titanates. This analysis indicated the conditions of theoretical chemical equilibrium to obtain potassium titanates. Thus, in the case of K_2_Ti_6_O_13_ (PHT), it could be obtained from 1100 °C. However during the experimental stage, the stoichiometric relationship in each system must be taken into account since according to Zaremba and Witowska, Bao et al. and Yaxin et al. [13,20,21,22], there is a strong dependence between the initial TiO_2_ content and the final product, observed in the corresponding equilibrium phase diagram reported by Bao et al. [20].
K_2_CO_3_ + TiO_2_ = K_2_TiO_3_ + CO_2_(g)       (Reaction 1)
K_2_CO_3_ + 2TiO_2_ = K_2_Ti_2_O_5_ + CO_2_(g)       (Reaction 2)
K_2_CO_3_ + 4TiO_2_ = K_2_Ti_4_O_9_ + CO_2_(g)       (Reaction 3)
K_2_CO_3_ + 6TiO_2_ = K_2_Ti_6_O_13_ + CO_2_(g)       (Reaction 4)

## 2. Potassium Hexatitanate

PHT is a material that has generated a lot of interest because of its economic potential. It is a fibrous material which has high mechanical strength, good thermal durability, chemical resistance, dispersion ability and is cheaper and softer than other reinforcements such as silicon carbide (SiC), so the composites reinforced with PHT can be easily machined with ordinary tools [23]. Besides, to understand and improve the properties of materials containing PHT, it is important to know their intrinsic physical properties; in this sense, the following sections resume the most relevant information generated by several researchers.

### 2.1. Structure

PHT crystallizes in the monoclinic system with space group C2/m; it has a tunnel-like structure formed by titania octahedra enclosing the atoms of K [24,25]; K_2_Ti_6_O_13_ crystals include two variants of the chemical formula (K_2_O.6TiO_2_ and K_2_Ti_6_O_13_) and a content of 42 atoms of which four are potassium (K), 12 titanium (Ti) and 26 oxygen (O). Likewise, the growth axes were mainly along the [010] direction [1,26].

The International Centre for Diffraction Data (ICDD) has reported two powder diffraction files (PDF) for the PHT: 40-0403 and 74-0275, which are included in the Inorganic Crystal Structure Database (ICSD); there is a slight variation in the network parameters. Table 1 presents the lattice parameters for each file and those obtained by Manyu et al. calculated using the Cambridge Serial Total Energy Package (CASTEP) [25].

Inside crystalline structure of the PHT, the interactions between the Ti and O are stronger than those between the K and O, while those between K and Ti almost do not exist, promoting TiO_6_ octahedra formation, which are joined by the corners or edges forming zigzag ribbons. Each one is joined above and below to similar groups constructing tunnels. The potassium ions in this material are enclosed or trapped into the structure, so they are isolated from the environment. This arrangement makes the material more resistant to chemical attack, compared with potassium di/tetra-titanate; meanwhile, it prevents potassium ions’ escape or exchange by other cations (Figure 2). This type of structure promotes whisker or fiber crystalline growth and provides the characteristics for different applications such as reinforcement agents, thermal insulators, photocatalyst, etc. [13,27,28,29,30,31].

Besides, it also has been reported that PHT prepared both hydrothermal and solid-state conditions presents similar absorption bands observed in the FTIR, which are at 457 cm^−1^, 500 cm^−1^, 713 cm^−1^, and 760 cm^−1^; the frequency at 713 cm^−1^ corresponds to TiO_6_ group and that at 470 −510 cm^−1^ corresponds to Ti–O [32].

### 2.2. Properties

In general, properties of materials depend on their microstructures and synthesis methods; in this sense, PHT has morphology of fibers or whiskers, and it plays a key role in its varied applications [25]. Some of the most important physical properties of PHT are density, hardness, and melting point according to applications that require better high thermal stability and resistance—for instance, as reinforced composite materials of low density and high thermal as aluminum alloys. Instead, silicon carbide is the commonly used material for its high mechanical strength and low thermal expansion coefficient; however, it has elevated cost, making it unviable for many industrial applications. Therefore, in terms of economic considerations, PHT is cheaper and softer than silicon carbide, so the resulting composites are easily machined with ordinary tools. It is also important to mention that PHT whiskers are very effective in improving the mechanical properties and thermal stability of thermoplastic materials [23].

Today, ceramic fibers used for performed composites must have low density, high resistance and stiffness; therefore, the strength and Young´s modulus of the fibers are features aimed for. In this context, previous works showed that the resistance to the fracture measured in the ceramic materials is substantially smaller than that predicted by the theory, based on experimental bonding forces between atoms. This conclusion can be explained by the presence of small defects in the material, which act as stress accumulator points. Nonetheless, the ceramic fibers exhibit competitive properties, and in the cases of the composites worked at high temperatures, the fibers should have a high melting point [33].

When the bulk modulus of PHT is calculated, it showed high resistance to volume change due to applied stress. Moreover, the cutting-effort modulus is a measurement of reversible deformation in shear stress, while the larger is the shear modulus; therefore, the more pronounced directional bonding is between atoms. Additionally, Young´s modulus provides a measure of the stiffness of a solid, and the high calculated value (122.5 GPa) proves that it is a very rigid material [26]. Similarly, the optical properties of the PHT for catalytic applications were investigated, e.g., in photocatalytic degradation, photocatalytic water splitting, and steam reforming of methane for hydrogen production. Researchers determined (theoretical and experimentally) the bandgap value of the PHT and the results showed that this material is a wide-broad semiconductor, and it presents a bandgap similar to the TiO_2_ [27,31,34,35,36,37].

Table 2 summarizes the physical and mechanical properties of PHT registered in literature, while Table 3 shows several bandgap data where the values vary slightly according to the method of synthesis.

Otherwise, thermal stability is a relevant intrinsic feature, described as the material ability to withstand transformation and/or decomposition, occasioned by prolonged exposure to high temperature. This property is determined due to the intrinsic crystalline structure and influenced by the heating conditions as a heating atmosphere and the temperature [45].

In relation with the stability, it is well known that ceramic materials are formed by combinations of metallic and non-metallic elements and are highly resistant to corrosion in diverse media; and its stability can be chemical (related to the dissolution of material or corrosion) or structural or dimensional (measured as the capacity to withstand loads at a given temperature) [46]. As an example, it is reported that PHT has high thermal stability, which decreases slightly at around 1200 °C; its calculated enthalpy of formation (−6035 kJ/mol) and cohesive energy (−137.4502 eV/f.u.) reveal high structural stability [25].

### 2.3. Methods of Synthesis

There are several methods that have been developed to synthesize PHT; the more known and used are calcination (and calcination with slow cooling), hydrothermal reaction, flux growth (or flux evaporation), ionic exchange and sol-gel synthesis [13,30], although this material also has been produced by other methods such as microwave-assisted, molten salts and self-propagating high-temperature synthesis [14,17,47,48]. Some of the most common methodologies are described below.

#### 2.3.1. Calcination (Solid-State Reaction)

The procedure that implies the direct heating of solids is also known as a reaction in solid-state, ceramic or calcination method. This is the simplest and the most common method to prepare solids; it involves heating together two or more non-volatile solids, and generating a product from their reaction. This method is widely used in the industry as well as in the laboratory and can be used to synthesize a broad range of materials [49].

The synthesis of PHT has been reported by calcination method and calcination with slow cooling. As an example, Li et al. used a homogeneous mixture of KF and TiO_2_ (anatase or amorphous gel) with a reason in weight of 2:1, which was treated thermally at 720 °C for 4 h. Subsequently, this mixture was washed with distilled water three times to produce an intermediate product; then, it was heated in boiling water for 5 h, and then, dried and re-heated at 800 °C for 1 h, obtaining K_2_Ti_6_O_13_ whiskers [27].

Likewise, Bao N. et al. reported the synthesis of PHT from a mixture of K_2_CO_3_ (reactive grade) with TiO_2_·H_2_O (previously obtained by hydrolysis of titanil sulfate in boiling water with vigorous agitation), using a molar ratio TiO_2_/K_2_O of 3.0. Initially, the mixture is dried in a vacuum stove at 90 °C for 10 h, after time, the mixture is calcinated at 1080 °C for 0.25 h to produce the K_2_Ti_6_O_13_ [50].

It has been also reported the application of a method of calcination with slow cooling, which involves a milling with ethanol of K_2_CO_3_ and TiO_2_ with a molar ratio K_2_O/TiO_2_ of 1:3 for 24 h and subsequently passing to a process of drying the sample obtained. Thereafter, the mixture is heated in platinum crucible at 1150 °C for 6 h; the cooling of the sample is gradually carried out to a temperature of 950 °C at a rate of 16 °C/h and then tempered in water. Finally, the sample is treated in boiling water for 4 h and later reheated at 1000 °C for 1 h, resulting in clean PHT fibers with a length that exceeds 100 µm [51].

Recently, Li M. et al. have reported the facile synthesis of PHT at temperatures below 800 °C. K_2_CO_3_, TiO_2_ and carbon black were mixed by ballmilling with ethanol for 3 h and then dried in at 80 °C for 10 h. The powders were mixed with 3 wt % polyvinyl alcohol and then pressed to obtain disks with 20 mm in diameter and 10 mm in thickness. Posteriorly, samples were calcined from 500 to 1100 °C for 3 h. With this methodology, a mixture of PHT and TiO_2_ (anatase phase) well crystallized micrometric whiskers were obtained from 800 °C [52].

It can be said that despite its extensive use, calcination method has some disadvantages: high temperature is required in the synthesis, the reactions are usually slow; the reactants should be finely grounded to achieve a homogeneous mixture and the products are often not homogeneous both in composition and particle size, which generally is micrometric.

#### 2.3.2. Hydrothermal Reaction

The hydrothermal method consists of heating reagents in a closed container (autoclave) with water. Between the components of autoclave, there is a thick stainless steel plate that resists high pressures and has all the respective safety valves. This container might be coated with non-reactive materials, such as noble metals, quartz or Teflon. When the autoclave is heated, the pressure increases, but the water inside remains liquid above its normal boiling point (100 °C), which is known as “superheated water”. For that reason, the hydrothermal conditions are defined as the pressure overhead the atmospheric and the temperature over the boiling point but not superior to other methods of synthesis [49].

Several authors reported the synthesis of PHT by the hydrothermal reaction, using different starting reagents and synthesis conditions. Thus, Meng et al. report experiments with the use of 2 g TiO_2_ powder dissolved in 50 mL of aqueous solution of KOH 15 M, stirred during 20 min and then introduced in steel autoclave with Teflon coating, using a temperature of 180 °C for 4 days. The final product is then filtered and washed with deionized water [41]. Moreover, Song et al. use a similar methodology to the one described above. Varying only the starting material, they used titanium tetrachloride and the time of permanence in the autoclave was 2 days [53]. In the same way, Masaki et al. used Ti powder and varied the temperature conditions from 150 to 350 °C and gave a permanence time in the autoclave of 2 h [7].

Likewise, Zaremba et al. reported the synthesis of PHT from a mixture of potassium hydroxide and titanium tetraisopropoxide as raw materials with a molar ratio of 1:2; both reagents were agitated for 30 min (an hour before), previous to heating in an autoclave, using a nickel tube. In this research, varied temperatures (350–450 °C) and time (2.5–25 h) were worked, and subsequently, the product was allowed to cool slowly in the autoclave followed by a filtrate, a wash with distilled water and ethanol; finally, it was dried at 50 °C for 24 h. The fibers obtained are stable, long, thin and with large surface area [13].

Moreover, Hakuta et al. [30], described the synthesis of nanowires of PHT by this route. In the approach, two starting solutions were prepared by dissolving titanium hydroxide and potassium hydroxide within distilled water, with a titanium ion concentration set to 0.02 M. Both solutions, were fed separately by a high-pressure pump, and then, they were mixed at the mixing tee and finally fed to the reactor; the next step consisted of adding preheated water, an elevated temperature reaction from 300 to 420 °C and setting the pressure to 30 MPa. The powder was dried in an oven at 60 °C for 24 h, with crystalline particles with 10 nm in width and length ranging from 500 to 1000 nm.

In the hydrothermal method, particle morphology and size of the whiskers were able to be controlled with the modified experimental conditions during the process; however, it was demonstrated that leaching by water converts layered titanates into amorphous structures [54,55].

#### 2.3.3. Flux Growth

In this type of synthesis, crystallization takes place from a solution, either by cooling a hot saturated solution or when evaporation occurs. Many crystalline substances have been grown from a saturated solution in a solvent. Several techniques can be used to induce crystallization, and these include flux growth and evaporation of fluxes by heating. For solids such as oxides, which are very insoluble in water, it may be possible to dissolve them in borate, fluoride or even molten metal; in this case, solvents are known as fluxes because they reduce the melting temperature of the solute. The molten is cooled slowly until the crystals are formed, and then the flux is emptied or dissolved separately. This method has been successfully used for the synthesis of crystalline silicates, quartz, alumina, titanates among many others [49]. In this sense, several fluxes have been used to obtain PHT: for the systems of a single component PbO, K_2_MoO_4_, Bi_2_O_3_ and B_2_O_3_ is used; and for the systems of 2 or more components K_2_CO_3_-K_4_P_2_O_7_, K_2_CO_3_-V_2_O_3_ PbO-K_2_P_2_O_7_, Li_2_O-K_2_MoO_4_, KCl-KF, K_2_MoO_4_-MoO_3_, Na_2_O-K_2_O-B_2_O_3_ are commonly used [2,3,4,27,30,56]. Both morphology and particle size cannot be controlled.

Crystalline substances have been grown from a saturated solution in a solvent. The crystallization process takes place either by cooling a hot saturated solution or when evaporation occurs; additional techniques include flux growth and evaporation of fluxes by heating. For solids such as oxides, which are very insoluble in water, it is possible to dissolve them in borate, fluoride or even molten metal. In the last case, they are known as fluxes because they reduce the fusion temperature of solute. The molten is cooled slowly until the crystals are formed, and then, the flux is emptied or dissolved separately.

Thus, for example, in the synthesis of PHT long fibers using Li_2_O-K_2_MoO_4_ as flux, the raw materials K_2_CO_3_, Li_2_CO_3_, K_2_MoO_4_ and TiO_2_ reagent grade were used. Flux and raw materials were mixed and melted into a platinum crucible at 1200 °C for 4 h and then rapidly cooled at room temperature. Finally, the products were washed in hot water and dried at 110 °C for 1 day [3]. Likewise, Choy and Han [29] reported the synthesis using the same flux but different conditions— they used a flux to raw material mole ratio of 7:3. After calcination at 1150 °C for 4 h, it was slowly cooled down to 900 °C, posteriorly quenched in air and finally washed with hot distilled water and dried at 110 °C. The PHT fibers obtained had a cylindrical shape with very clean and smooth surface and uniform particle size of 1.5 mm length and 2–3 µm diameter.

Moreover, recently Ponce-Peña et al. reported the use of boric acid (H_3_BO_3_) as a flux to obtain PHT fibers. The methodology employed in this research consisted in preparing a mixture of (molar %) TiO_2_ (60), K_2_O (30) and B_2_O_3_ (10), which was melted at 1250 °C for 1 h and poured into water afterwards. The obtained material is dried, grounded in agate mortar (mesh-80, 180 μm), washed in hot water (95 °C) to remove the remaining flux and dried. Finally, the product is thermally treated between 900 and 1000 °C for 2 h. Figure 3 shows the PHT micrometric fibers obtained from this method [31].

#### 2.3.4. Ionic Exchange

The ion exchange process is a reversible exchange of ions between solid and liquid phases [57]. PHT may be produced by the ion exchange method, which was demonstrated by Um et al. [16,58]. The synthesis process is described as follows. First, KHTi_4_O_9_ is produced by exchange, half of the K ions in K_2_Ti_4_O_9_ with H^+^ ions using 0.005M HCl solution; second, KHTi_4_O_9_ is thermally treated at 250–500 °C for 3 h, resulting in K_2_Ti_8_O_17_ formation; third, K_2_Ti_8_O_17_ is given heat treatment above 600 °C to produce a mixture of K_2_Ti_6_O_13_ and TiO_2_.

Liu et al. also reported the synthesis of PHT fibers using potassium titanate (K_4_Ti_3_O_8_) as a starting material. K_4_Ti_3_O_8_ was produced by the treatment of TiO_2_ with 80 wt % concentrated KOH solution at 220 °C for 2 h, under normal pressure. Later, 0.25 g of K_4_Ti_3_O_8_ is shaken with 0.025 dm^3^ (25 cm^3^) solutions (containing an appropriate amount of 0.1 mol/L HCl), for 3 days at 25 °C, to obtain a solid (hydrolysis intermediate), which was filtered, washed and dried at room temperature in a desiccator. Finally, with thermal treatment at 610 °C for 2 h, the hydrolysis intermediates were transformed into well-crystallized PHT fibers [59].

#### 2.3.5. Sol-Gel Method

Among the techniques above-mentioned, the sol-gel route offers some specific advantages, e.g., from the use of chemically homogeneous precursor is possible to ensure atomic-level mixing of reagents; moreover, lower processing temperatures and shorter synthesis times are possible; furthermore, this method should enable greater control over particle morphology and size [60].

In this way, sol-gel method has been applied to obtain PHT. For example, Jung and Shul [28] reported obtaining this material using the following methodology: Initially, the precursor of potassium, water and catalyst were dissolved in 0.5 mL EtOH; next, the precursor of Titanium dissolved in EtOH remnants was added to the first solution to ensure a homogeneous solution; after, the reaction proceeded by hydrolysis. Subsequently, the solution was aged at room temperature from one to seven days to ensure the freezing of the structure. Then, the sample was dried and calcined to 700 °C to produce the potassium hexatitanate with a surface area between 30 and 40 m^2^/g.

Moreover, Qian et al. prepared PHT nanofilms following the next steps: First, potassium acetate is dissolved in acetic acid (solution 1); second, a stoichiometric molar quantity of Ti(n-OC_4_H_9_)_4_ is mixed with acetylacetone (solution 2); third, solution 1 was dropped into solution 2, with continuous stirring during 2 h. The final solution is heated at 70 °C to form a gel, which is grounded and calcined at different temperatures for 2 h [61].

As above-mentioned, this method permits the control of particle size from micro to nanometric. On this subject, Kang et al. [62] reported the synthesis of PHT nanorods, using a reactive mixture of CH_3_OK (95%) and Ti(OC_2_H_5_)_4_ (technical grade) in 20 mL of C_2_H_5_OH, with a molar ratio of 1:1 to 1:2.8 (C_2_H_5_OH to Ti(OC_2_H_5_)_4_). To control the competitive reactions hydrolysis and condensation in the medium, distilled water and HCl (35%) were used, and the pH remained in 7. The mixture was stirred at 40 °C for 2 h to produce the Sol, which aged during 100 h at room temperature. Posteriorly, the sol was dried for 48 h at 100 °C to obtain a Xerogel. Finally, it is calcined at temperatures from 800 to 1050 °C for 3 h. The K_2_Ti_6_O_13_ nanorods collected have a diameter roughly below 60 nm.

#### 2.3.6. Comparison of Synthesis Methodologies

The above described synthesis methodologies have advantages and disadvantages. For comparison, some of these characteristics are presented in Table 4.

### 2.4. Applications

Principal applications of PHT fibers are a function of their properties, mainly physical, mechanical, friction and optical; this material has been used as reinforcing agent (in polymers, ceramics and metals) and photocatalyst (to produce hydrogen and decomposition of organic contaminants). In this area, it is well known that most fiber-reinforced compounds achieve better resistance to fatigue, better stiffness and better strength-weight ratio by incorporating tough, rigid but fragile fibers in a softer, more ductile matrix. The microstructure of fiber reinforced composite materials is made up of the fiber, the matrix and an interface. In the interface the physical, mechanical and chemical properties differ from those of the original fiber and the matrix; some features occurring in this region are variable crosslink density and molecular weight, trans-crystallinity, impurities, sizing, voids, fiber surface chemistry, fiber topography and morphology. The interface forms the fiber/matrix bonding, so the matrix transfers the force to the fibers, which support most of the applied force. Compound strength can be high at room and elevated temperatures [63,64]. Fiber reinforced composites can be classified into four groups according to their matrices: metal matrix composites (MMCs), ceramic matrix composites (CMCs), carbon/carbon composites (C/C) and polymer matrix composites (PMCs) or polymeric composites [63]. In this sense, is interesting to mention that PHT fibers have been used to reinforcing polymers, ceramics and metal alloys to obtain composites, which have a wide application (in automotive, money paper, refractories, etc.) because of their price, which is lower than other reinforcing agents such as silicon carbide, silicon nitride, aluminum borate, among others [23,65,66,67].

On the other hand, when it comes to talking about friction materials, it is known that chrysotile asbestos fibers have been conventionally used for brake devices. However, these materials have low thermal stability and their coefficient of friction rapidly decreases at relatively low temperatures to undergo fading, while the friction material markedly wears away at high temperatures, in addition to being a cancerous and toxic agent, and asbestos has been gradually substituted by non-asbestos inorganic fiber [68,69,70]. In this context, PHT fibers showed low pulmonary toxicity in studies carried out in rats using a single 2-mg dose per rat [71]; the above-mentioned makes this material promissory for engineering applications. The next subsections describe several uses of the PHT reported so far.

#### 2.4.1. PHT as Reinforcing Agent

The principal use of PHT fibers/whiskers is in the field of reinforcing polymers, there are several works that describe the properties and uses of these composites. However, it is important to consider for short fiber polymer composites two significant aspects which have great influence on the properties of composites: The aspect ratio of reinforcement material and the interfacial adhesion between the polymer matrices and reinforcements [72]. In addition to the above, it is important to mention that PHT fibers often exhibits easy aggregation and poor dispersion in polymer matrix; moreover, the large surface energy difference between hydrophilic PHT and hydrophobic polymer could be the principal cause of the poor interfacial interaction, which leads to the decrease of the composite’s performance [73,74]. For the above-mentioned reasons, it is necessary to use surface-treated fibers with a coupling agent, in order to improve both the homogenous distribution and the adhesion to the matrix. Examples of employed coupling agents are silanes such as vinylsilane, aminosilane, epoxysilane, methacryloxysilane and mercaptoxysilane and titanates such as isopropyltriisostearoyl titanate and di(dioctylpyrophosphate)- ethylene titanate, tetrabutyl orthotitanate, etc. [38,68]. Table 5 summarizes the main findings carried out by several researchers on the synthesis of polymeric composites reinforced with PHT fibers/whiskers, although, its use also has been reported to reinforce keratin, paper currency and scrap tire [70,75,76,77].

PHT also has been used to reinforce alloys, principally light aluminum and magnesium alloys [65]; in this area, several research works can be cited in which PHT fibers/whiskers have increased mechanical and wear resistance and diminished the coefficient of friction and thermal expansion of the composites [23,66,67,78,79,80]. Likewise, PHT fibers/whiskers are effective for the prevention of fading because of their high heat resistance. Mohs’ hardness of about 4 is therefore less likely to abrade the adjoining materials and is useful, for example, for preventing abnormal actuation of brakes since it has less hygroscopicity, and is not reactive with water [68]. However, as mentioned above, the properties of the interface in composites depend on the interface bonding between the fiber and matrix; a slight reaction may enhance the interface keying and the strength of composites, but a severe reaction could destroy the structural integrity and the reinforcing effect of the reinforcements, resulting in a reduction of the strength of the composite [70]. Thus for example, the PHT whiskers are destroyed by their reaction with aluminum and diffusion of the K element into aluminum, with loss of the reinforcing function; but fortunately, the modifications of the surface reinforcements can improve the interface between the reinforcement and matrix [78]. Generally, reinforcement surface treatment consist in recovering it with Al_2_O_3_ by sol-gel process, using a isopropoxide as the precursor which is hydrolyzed in water and alcohol to produce a clear “sol”, and then, the whiskers are dipped into the sol for several minutes and finally pyrolyzed to obtain the oxide coating; with this, the fiber/whisker wettability is improved [78,80,81]. Generally, a volume fraction of the PHT fibers used is between 5% and 40%. For a smaller amount, their effectiveness in reinforcing the matrix alloy is very small, and if it is more than 40%, lesser improvement is achieved in strength as the volume fraction of the PHT fibers increases. In addition, an increasing use of the potassium hexatitanate fibers results in a higher cost of composite materials [65]. Most of these obtained composites are destined for the automotive industry for use in parts, such as all types of internal combustion engines as pistons, cylinder blocks and heads.

On the other hand, in the case of ceramics reinforcement, there are a few works which describe several composites with PHT fibers/whiskers. Generally, these composites are obtained by in situ growing of potassium hexatitanate using potassium polytitanates as starting material. Thus, for example, composites have been reported in the systems of alumosilicate-potassium polytitanate, potassium titanate-metalurgical slag, potassium polytitanate-glass SiO_2_-B_2_O_3_-R_2_O-Al_2_O_3_, which could be used as high strength ceramics in applications related to structural, refractory, frictional and heavy metal adsorption materials [82,83,84,85]. Likewise, recently, hydroxyapatite (HA) based composites are attracting considerable interest for use as engineering components such as bearings and mechanical seals since, compared with metals, HA-based composites show a lower density and substantially great corrosion and heat resistance [86].

#### 2.4.2. PHT as Photocatalyst

Since photocatalytic solar energy conversion to hydrogen is an environmentally benign technology, the development of efficient photocatalysts has been extensively studied all over the world [36]. In this sense, the application of the PHT as a catalyst is principally focused on the production of hydrogen from water or methanol splitting and degradation of organic compounds. Thus, for example, Yoshida et al. [34] studied the photocatalytic activity of PHT (obtained by flux growth) in the reaction for water splitting or H_2_/O_2_ evolution from aqueous solution; the samples prepared with PHT and rhodium as co-catalyst showed photocatalytic activity achieving to generate up to 0.12 μmol/min of H_2_. Likewise, Escobedo Bretado et al. evaluated the photocatalytic activity of PHT fibers (obtained by flux growth), and the results showed that the maximum amount of hydrogen achieved was 2387 μmol of H_2_/g_cat_ in 8 h of irradiation with UV light [37]. Yahya et al. too reported a H_2_ evolution rate from H_2_O of up to 47 μmol/h in samples prepared with the loaded ruthenium oxide on PHT [97]. Likewise, Hakuta et al. [31] report the use of PHT nanowires (prepared by hydrothermal method) with large surface area (up to 219.8 m^2^/g) and high photocatalytic activity, in the reaction of photolysis of methane; the amount of hydrogen generated by the use of this material was approximately 0.2 μmol in a reaction time of 8 hours. Moreover, Guan et al. [98] have reported the photoreduction of CO_2_ with water into methanol under concentrated sunlight using a Pt-loaded PHT photocatalyst or a composite catalyst in which the Pt-PHT photocatalyst was combined with a CO_2_ hydrogenation catalyst of Cu/ZnO. On the other hand, PHT also has been used for degradation of a methylene blue (MB). According with Takaya S. et al. [26], the results showed that PHT with nano-needle crystals had high photocatalytic activity and the degradation of MB solution by photocatalytic activity of PHT at 20 h, was about three times higher than those of TiO_2_ thin films.

#### 2.4.3. Other Applications

Recently, titanates have been introduced to use as biomaterials. There are a few studies in which in vitro bioactivity was investigated. It has been reported that potassium titanates are biocompatible, and potassium titanate nanorod arrays can induce the formation of apatite after 4 days in SBF (simulated body fluid) solution. The bioactive PHT whiskers have good biocompatibility and are potential candidates as reinforcing agents to improve the mechanical properties of calcium phosphate ceramics or cements, in order to apply them in load conditions [99,100,101,102]. Another application was established by Gonzáles-Lozano et al. [103]. These authors reported the obtention of a glass-ceramic with PHT, which can be promissory as solid lubricant; the data on friction behavior were obtained using a rheometer with a cylindrical cavity in which the material was deposited. Results showed that this glass ceramic had a friction force 0.18 ± 0.06 N, under a normal load of 20 N at room temperature (23 °C), which was lower than that achieved by the reference glassy solid lubricant.

## 3. Conclusions

This review covers much and interesting information about the structure, synthesis, properties and applications of the PHT. The structure and properties of the PHT makes it a multifunctional material that can be applied mainly in the reinforcing of metals, polymers and ceramics. However, in recent years, many researchers have focused on the use of the PHT as Photocatalyst in the reactions for the production of hydrogen from water and methane, in the degradation of organic contaminants (e.g., orange G, methylene blue and amoxicillin) and in the photoreduction of CO_2_; for these applications a small particle size is required. In this sense, nanoparticles obtained by hydrothermal or sol-gel synthesis could be used. Other promising but less investigated applications are in the field of biomaterials, adsorbents for heavy metals and solid lubricants. It is expected that much of the information concerning PHT is spread with this research and is useful for the design of new materials, (e.g., the doping of PHT for the enhancement of photocatalytic, ion exchanging and biocompatibility properties), as well as in the development of new applications in various fields of science, taking advantage of their inherent properties and great potential of this material.

## Figures and Tables

**Figure 1 materials-12-04132-f001:**
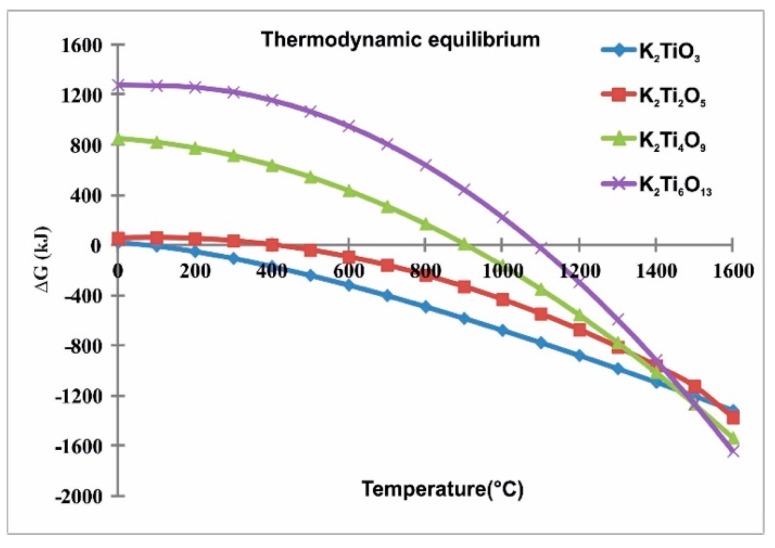
Thermodynamic analysis to obtain potassium titanates from K_2_CO_3_-TiO_2_ system; this diagram was obtained using the software HSC Chemistry 7.0.

**Figure 2 materials-12-04132-f002:**
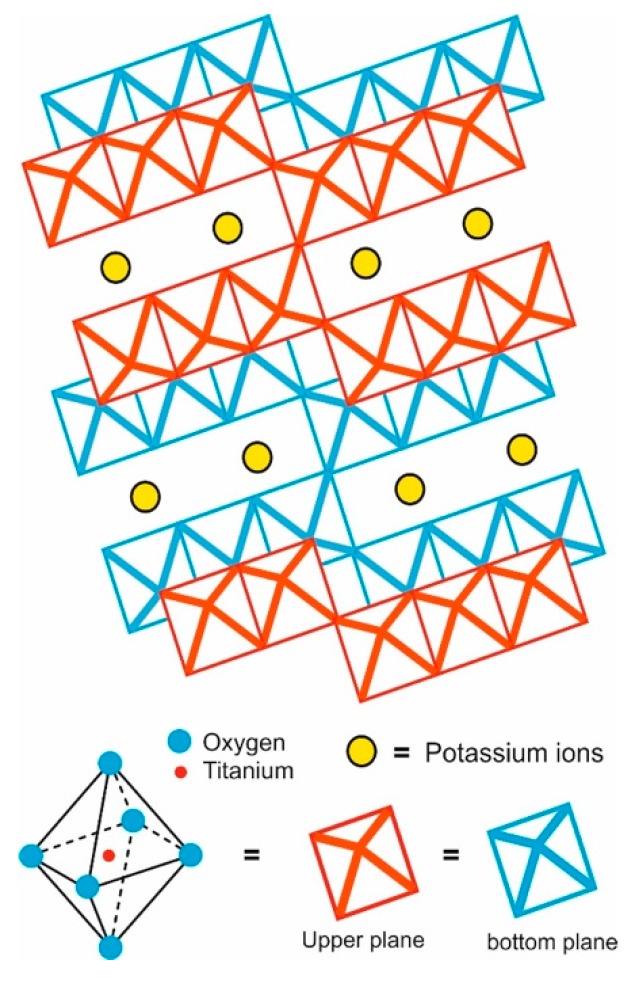
Representation of potassium hexatitanate structure.

**Figure 3 materials-12-04132-f003:**
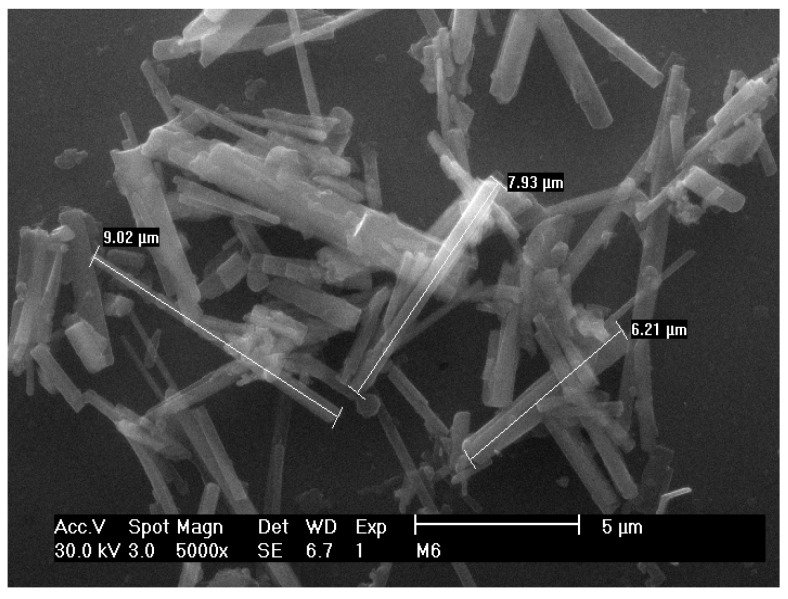
PHT fibers obtained by flux growth method.

**Table 1 materials-12-04132-t001:** Lattice parameters for the PHT.

PDF	Parameters			
	a (Å)	b (Å)	c (Å)	β (°)
PDF 40-0403	15.593	3.796	9.108	99.78
PDF 74-0275	15.582	3.82	9.112	99.764
Calculated [25]	16.1529	3.7801	9.3388	101.1878

**Table 2 materials-12-04132-t002:** K_2_Ti_6_O_13_ properties reported by several authors [21,38,39].

Property	Value
Density	3.2–3.3 g/cm^3^
Melting point	1370 °C
Softening point	1200 °C
Mohs hardness	4
pH of water slurry	7–8
Thermal expansion coefficient	6.8 × 10^−6^ K^−1^
Specific heat	920 J/kgK
Tensile strength	7 Gpa
Tensile modulus	280 Gpa

**Table 3 materials-12-04132-t003:** Potassium hexatitanate (PHT) band gap reported by several authors.

Synthesis Method	Band-Gap (eV)	Reference
Hydrothermal synthesis	3.45	Du G.H. et al. [40]
Hydrothermal synthesis	3.3	Meng X. et al. [41]
Solid-state reaction	3.52	Yoshida H. et al. [34]
Solid-state reaction	3.06	Siddiqui M.A. et al. [42]
Flux growth	3.3	Ponce-Peña P. et al. [31]
Sol-gel synthesis	3.48	Siddiqui M.A. et al. [42]
Low-temperature synthesis	3.47	Li J. et al. [43]
Sonochemical method	3.42	Sehati S. and Entezari M.H. [44]

**Table 4 materials-12-04132-t004:** Advantages and disadvantages of the most common synthesis methods for PHT obtaining.

Synthesis Method	Advantages	Disadvantages
Calcination	Facile and economic synthesis. Suitable for large-scale production.	Require high temperatures. Reagents finely grounded. Difficult particle size control.
Hydrothermal reaction	Low temperature synthesis. Easy morphology and size (micro/nanometric) particle control. Product with high purity	Long synthesis time. PHT structure can be amorphized by supercritical water. Is not suitable for mass production.
Flux growth	Can be use a great variety of fluxes. Size (micrometric) and morphology can be controlled.	Fluxes can corrode the crucibles. Many stages are required.
Ionic exchange	Low temperature synthesis. Easy morphology and size (micro/nanometric) particle control.	Many stages are required. Long synthesis time. Is not suitable for mass production.
Sol-gel	Low temperature synthesis. Size (micrometric) and morphology can be controlled. Product with high purity.	Expensive starting materials Many stages are required.

**Table 5 materials-12-04132-t005:** Principal polymer-K_2_Ti_6_O_13_ composites and their characteristics.

Polymeric Matrix	K_2_Ti_6_O_13_ Content (wt.%)	Others Reinforcing Agents Used	Used Coupling Agent	Properties	Composite Application	Processing Method	Reference
**PP**	10	Ramier Fiber	Silane	Tensile strength (30 Mpa). Bending Strength (63.5 Mpa). Compressive strength (51.5 Mpa). Impact strength (8.6 KJ/m)	Automotive and aircraft	Extrude molded	Long C.-G. [87]
**PP**	5–35		Tetrabutyl orthotitanate	15% K_2_Ti_6_O_13_ composite Tensile strength (32.42 Mpa). Young´s module (2.833 Gpa). Longitudinal impact strength (32 J/m). Transverse impact strength (30.8 J/m)	Load bearing applications	Twin-screw extruded follow by injection molded	Tjong S.C. [88]
**PA-6**	5–35	-	Tetrabutyl orthotitanate	25% K_2_Ti_6_O_13_ composite Tensile strength (69 Mpa). Young´s module (2.65 Gpa). Longitudinal impact strength (59.5 J/m). Transverse impact strength (45 J/m)	Automobile parts	Injection molded	Tjong S.C. [89]
**PA-6**	5–15	-	Propyltrimethoxy-silane	15% K_2_Ti_6_O_13_ composite Tensile strength (86.03 Mpa). Tensile Modulus (31.96 Gpa). Impact strength (7.97 ln-lbs)	-	In situ polymerization	Yuchun et al. [90]
**ER**	0–7.5	Glass fibers	-	7.5% K_2_Ti_6_O_13_ composite Density (1.69 g/cm^3^). Rockwell M Hardness (99). Tensile strength (247 Mpa). Flexural strength (274 Gpa). Impact strength (1.86 J/mm) FC range (0.430-0.451) Specific wear rate (1X10^-5^ mm^3^/Nm	-	Vacuum molded	Sudheer et al. [91]
**PR**	10–15	Barite (BaSO_4_)GraphiteAlumino-silicate fibersAramid fibers	-	15% K_2_Ti_6_O_13_ Density 2.02 Tensile strength (9.4 Mpa). Flexural strength (44.71 Mpa) Impact strength (0.2 KJ/m^2^) Elongation (1.31%)	Braking applications	Mixing follow by compression molding	Kumar et al. [92]
**PEEK**	10–30	-	-	30% K_2_Ti_6_O_13_ compounded rheometer composite Tensile strength (125 Mpa).	Chemical, mechanical, aeronautic, electronic and nuclear industries	Twin-screw extruder or Torque rheometer	Zhuang et al [72]
**PEEK**	15	Carbon fibers	-	15% K_2_Ti_6_O_13_ composites Water absorption (0.71%), FC (0.01) Wear rate (9.2 × 10^−9^ mm^3^/Nm, at 15Mpa)	Chemical, mechanical, aeronautic, electronic and nuclear industries	Injection molded	Xie et al. [93]
**PTFE**	0–40	-	Aminosilane	20% K_2_Ti_6_O_13_ composites FC (0.127) Wear rate (8.38 × 10^−10^ cm^3^/Nm) at 100N and 1.4 m/s) Heat of fusion (44.487 J/g)	Bearing and sealing materials	Compression molding	Feng et al. [94,95]
**PTFE/PEEK**	0–30	-	n-octodecyl-triclorosilane	10% K_2_Ti_6_O_13_ composites FC (0.125) Wear rate (3.37 × 10^−5^ m^3^/Nm) at 100N and 1.4 m/s Tensile strength (82.8 Mpa), Flexural strength (113.1 Mpa), Impact strength (15.9 KJ/m^2^)	Chemical, mechanical, aeronautic, electronic and nuclear industries	High temperature compression moulding	Huaiyuan et al. [74]
**PC**	5–25	-	Methyl-trimethoxy silane or Tetrabutyl orthotitanate	10% K_2_Ti_6_O_13_ composites Tensile strength (40Mpa)	Engineering thermoplastic	single-screw extruded follow by injection moulded	Jiang and Tjong [67]
**PP/LCP**	0–35	-	Tetrabutyl orthotitanate	35% K_2_Ti_6_O_13_ composites Tensile strength (45 Mpa) Young´s module (6 Gpa), Longitudinal impact strength (4.2 KJ/m^2^), Transverse impact strength (2.7 KJ/m^2^)	-	Extruded follow by injection moulded	Tjong and Meng [38]
**PP/PA-6**	0–40 (phr)		γ-methacryloxy-propyltrimethoxy silane and γ-aminopropyl-triethoxy silane	20 phr K_2_Ti_6_O_13_ composites ε_r_ (3.43) Tan δ (5.7 × 10^−3^) Tensile strength (75 Mpa) Impact strength (4.4 KJ/m^2^)	High performance insulating materials for electric applications	Torque rheometer	Yu et al. [96]

Polypropilene, PP; poly ether ether ketone, PEEK; polytetrafluoroethylene, PTFE; polycarbonate, PC; liquid crystalline polymer, LCP; polyamide, PA; epoxy resin, ER; phenolic resin, PR; friction coefficient, FC; parts per hundred part of resin, phr.

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
