# Peer review of "Crystalline Structure, Synthesis, Properties and Applications of Potassium Hexatitanate: A Review"

_materials, 2019, doi:10.3390/ma12244132_

Round 1
Reviewer 1 Report
Materials 650228
Crystalline structure, synthesis, properties and applications of potassium hexatitanate: a review
The review presents information on the crystalline structure, synthesis, properties and applications of potassium hexatitanate. The review is well documented, with a number of 104 references.
I have only few observations:
The description of the generalities of synthesis methods is too long (i.e. too much generalities for hydrothermal methods, L 196 – 208, and for flux growth method). The numbering of titles and subtitles must be corrected. They are numbered as follows:“2.1. Properties
2.1. Methods of synthesis
2.1.1. Calcination (solid-state reaction)
2.1.1. Hydrothermal reaction
2.1.1. Flux growth
2.1.1. Ionic exchange
2.1.1. Sol-gel method
2.1. Applications
2.1.1. PHT as reinforcing agent
2.1.1. PHT as photocatalyst
2.1.1. Other applications”
Author Response
The description of the synthesis methods was shortened.
The numbering of titles and subtitles was corrected.
Additionally english was improved.

Reviewer 2 Report
1#: The structure of the paper is not clear, such as “2.1. Properties”, “2.1. Methods of synthesis”, “2.1.1. Calcination (solid-state reaction)”, “2.1.1. Hydrothermal reaction”, “2.1.1. Flux growth”, “2.1.1. Ionic exchange”, “2.1.1. Sol-gel method”, “2.1. Applications”, “2.1.1. PHT as reinforcing agent”, “2.1.1. PHT as photocatalyst”, “2.1.1. Other applications” and “2. Conclusions”.
2#: The first and second paragraphs in section 2.1.1.(Flux growth) are repeated.
3#: The citation of ref. 16 and 28 in page 4 are wrong. These references have nothing to do with and bandgap of PHT.
4#: The authors should compare the advantages and disadvantages of the preparation methods, rather than simply list them in section 2.1.(Methods of synthesis).
5#: The content of 2.1. Applications is mainly to copy the conclusions of the references, without the author's own evaluation and language. For example, the author should compare the properties of catalysts prepared under different conditions, and indicate which catalyst has the best application prospect in the content of 2.1.1. (PHT as photocatalyst)
6#: After summarizing the current situation of PHT research, the authors did not present any new ideas or comments. In the conclusion, the authors should put forward the prediction which is beneficial to the future research.
Reviewer 3 Report
The paper is generally well written and good review on the synthesis methods and applications of PHT. Nevertheless, some corrections should be applied prior to publication.
The merging of all crystallographic data mentioned in the text (lines 84-86, Tables 1 and 2) into one table would be advantageous from the aspect of better comparability.
The section numbering is not consistent and should be revised.
The length of sentences are often too long, and contains too much parts which makes the text hard readable, simplification would be advantageous.
There are some language mistakes e.g. not contol of ... in lines 248 or 260
There are some mistypings, e.g. polytinate instead of polytitanate in line 395
Author Response
Point 1. The paper is generally well written and good review on the synthesis methods and applications of PHT. Nevertheless, some corrections should be applied prior to publication.
Response 1. The document was revised and corrected, all corrections are yellow highlighted.
Point 2. The merging of all crystallographic data mentioned in the text (lines 84-86, Tables 1 and 2) into one table would be advantageous from the aspect of better comparability.
Response 2. Tables 1 and 2) were joined in only one table (Table 1).
Point 3. The section numbering is not consistent and should be revised.
Response 3. The numbering of titles and subtitles was corrected.
Point 4. The length of sentences are often too long, and contains too much parts which makes the text hard readable, simplification would be advantageous.
Point 5. There are some language mistakes e.g. not contol of ... in lines 248 or 260
Point 6. There are some mistypings, e.g. polytinate instead of polytitanate in line 395
Response 4, 5 and 6. The document was revised and corrected, all corrections are yellow highlighted.
Reviewer 4 Report
The authors present a detailed review about synthesis, structure, properties and application of the material potassium hexatitanate. It therefore fits into the scope of the Materials journal. The manuscript is clearly structured and the information is presented in an easily accessible way.
However, the topic is very specific and the material itself does not seem to attract much attention of the scientific community during the last decades. Therefore, I think that a review would mostly be interesting for technology and a small group of experts in science. Having this in mind it, the amount of very basic/fundamental information given in the beginning of some sections seems a little too much (e.g. in the sections "calcination" or " hydrothermal reaction").
Potentially, impact of this paper on technology and application (e.g. by the use as reinforcement fiber or friction material) may be given through the outline of synthesis routes and proper embedding into a matrix.
One issue is the level of English language that is presented in the manuscript. This includes missing words, incorrect usage of words, spelling or grammar and, maybe most obviously, repetition of a whole paragraph (2nd paragraph in "Flux growth"). This makes the manuscript seem immature even though a lot of research must have been done to compile the presented information. Just as examples, the plural of "octahedron" should be octahedra and I don't think that "fusion temperature" can be used to refer to "melting temperature". In some cases, language issues made it impossible to understand certain sentenses. I strongly recommend to have the text proofread by an english expert with some background in science.
Furthermore, I would suggest to include the results of recent (this and last years) research into the review. For instance, a very recent publication regarding the synthesis can be found here: 10.1007/s41779-019-00409-4
A few more articles can be found easily by searching the relevant databases.
Overall I would, in principle, endorse publication of this work in Materials after improving the readability of the text and updating the content with recent publications. For scientists and engineers beginning to work with this material, this review can turn out to be very useful.
Author Response
Point 1. However, the topic is very specific and the material itself does not seem to attract much attention of the scientific community during the last decades. Therefore, I think that a review would mostly be interesting for technology and a small group of experts in science. Having this in mind it, the amount of very basic/fundamental information given in the beginning of some sections seems a little too much (e.g. in the sections "calcination" or " hydrothermal reaction").
Response 1. With respect to this opinion, is observed that during the last decades PHT has had a great interest by researchers, principally for catalytic applications. Likewise, the authors consider that all information contained in this review is relevant.
Point 2. Potentially, impact of this paper on technology and application (e.g. by the use as reinforcement fiber or friction material) may be given through the outline of synthesis routes and proper embedding into a matrix.
Response 2. The authors consider that the way of presenting the processing routes and applications in different sections improves reading and understanding information.
Point 3. One issue is the level of English language that is presented in the manuscript. This includes missing words, incorrect usage of words, spelling or grammar and, maybe most obviously, repetition of a whole paragraph (2nd paragraph in "Flux growth"). This makes the manuscript seem immature even though a lot of research must have been done to compile the presented information. Just as examples, the plural of "octahedron" should be octahedra and I don't think that "fusion temperature" can be used to refer to "melting temperature". In some cases, language issues made it impossible to understand certain sentences. I strongly recommend to have the text proofread by an english expert with some background in science.
Response 3. The document was revised and corrected, all corrections are yellow highlighted.
Point 4. Furthermore, I would suggest to include the results of recent (this and last years) research into the review. For instance, a very recent publication regarding the synthesis can be found here: 10.1007/s41779-019-00409-4
Response 4. In the document a new reference 10.1007/s41779-019-00409-4 was added although the paper was published just a month ago.
Point 5. Overall I would, in principle, endorse publication of this work in Materials after improving the readability of the text and updating the content with recent publications. For scientists and engineers beginning to work with this material, this review can turn out to be very useful.
Response 5. The document was english revised and corrected. Authors consider that there are a good number of references (105), of which 69 are from the last two decades and 36 of them are from 2010 onwards.
Round 2
Reviewer 2 Report
The authors have addressed the minor points successfully. However, the article still lacks valuable ideas or comments.